# Epidemiological and Molecular Investigation of the Heater–Cooler Unit (HCU)-Related Outbreak of Invasive *Mycobacterium chimaera* Infection Occurred in Italy

**DOI:** 10.3390/microorganisms11092251

**Published:** 2023-09-07

**Authors:** Angela Cannas, Antonella Campanale, Daniela Minella, Francesco Messina, Ornella Butera, Carla Nisii, Antonio Mazzarelli, Carla Fontana, Lucia Lispi, Francesco Maraglino, Antonino Di Caro, Michela Sabbatucci

**Affiliations:** 1National Institute for Infectious Diseases Lazzaro Spallanzani IRCCS, 00149 Rome, Italy; angela.cannas@inmi.it (A.C.); francesco.messina@inmi.it (F.M.); ornella.butera@inmi.it (O.B.); antonio.mazzarelli@inmi.it (A.M.); carla.fontana@inmi.it (C.F.); 2Unit 5, Directorate General of Medical Devices and Pharmaceutical Service, Ministry of Health, 00144 Rome, Italy; a.campanale@sanita.it (A.C.); d.minella@sanita.it (D.M.); l.lispi@sanita.it (L.L.); 3Unit 5, Directorate General Health Prevention Communicable Diseases and International Prophylaxis, Ministry of Health, 00144 Rome, Italy; f.maraglino@sanita.it (F.M.); m.sabbatucci@sanita.it (M.S.); 4Department of Microbiology, Unicamillus International University of Medicine, 00131 Rome, Italy; antonino.dicaro@unicamillus.org; 5Department Infectious Diseases, Istituto Superiore di Sanità, 00161 Rome, Italy

**Keywords:** nontuberculous mycobacteria, heater–cooler units, device, molecular surveillance, WGS

## Abstract

Background: From 2013 onwards, a large outbreak of *Mycobacterium chimaera* (MC) invasive infection, which was correlated with the use of contaminated heater–cooler units (HCUs) during open chest surgery, was reported from all over the world. Here, we report the results of the epidemiological and molecular investigations conducted in Italy after the alarm raised about this epidemic event. Methods: MC strains isolated from patients or from HCU devices were characterized by genomic sequencing and molecular epidemiological analysis. Results: Through retrospective epidemiological analysis conducted between January 2010 and December 2022, 40 possible cases of patients infected with MC were identified. Thirty-six strains isolated from these patients were analysed by whole genome sequencing (WGS) and were found to belong to the genotypes 1.1 or 1.8, which are the genotypes correlated with the outbreak. Most of the cases presented with prosthetic valve endocarditis, vascular graft infection or disseminated infection. Among the cases found, there were 21 deaths. The same analysis was carried out on HCU devices. A total of 251 HCUs were found to be contaminated by MC; genotypes 1.1 or 1.8 were identified in 28 of those HCUs. Conclusions: To ensure patients’ safety and adequate follow-up, clinicians and general practitioners were made aware of the results and public health measures, and recommendations were issued to prevent further cases in the healthcare settings. The Italian Society of Cardiac Surgery performed a national survey to assess the incidence of HCU-related MC prosthetic infections in cardiac surgery. No cases were reported after HCU replacement or structural modification and disinfection and possibly safe allocation outside surgical rooms.

## 1. Introduction

*Mycobacterium chimaera* (MC) is a nontuberculous mycobacteria (NTM) belonging to the *M. avium* complex (MAC) group. Identified in 2004 for the first time [1], it is widespread in the environment (water and soil), and while it is generally harmless for healthy subjects, it poses a threat to vulnerable patients who may develop infections of varying severity affecting soft tissue, surgical wounds and the respiratory tract [2]. Disseminated infection or death represent the worst outcome, especially when diagnosis is delayed.

In recent years, a multicountry outbreak of invasive infections by MC, related to contaminated heater–cooler unit devices (HCUs) used during cardiac surgery procedures, has been reported [3]. HCUs are devices used to control the temperature of patients’ blood during open-heart surgery and extracorporeal circulation, and the aerosol produced during the surgical procedure was identified as the most likely source of contamination of the surgical site. Results of early investigations linked these invasive infections to HCUs from a single manufacturer (Livanova 3T HCDS, formerly Sorin, London, UK), and the most likely hypothesis is that the contamination occurred at the manufacturer’s site [4,5]. By 2018, over 120 cases had emerged in Europe, North America, Australia, Hong Kong [4].

Initial symptoms may be nonspecific and often misdiagnosed (i.e., as sarcoidosis, vasculitis) and may appear months to years after surgery, ranging from 6 weeks to as long as 7 years (median latency estimated around 15–17 months after surgery) [6,7]. Extra thoracic symptoms (i.e., osteoarticular manifestations, cytopenia, hemophagocytic syndrome, nephritis or hepatitis, chorioretinitis, encephalitis) may precede cardiac or vascular manifestations, while signs of cardiac infection may be absent and detected only at surgery or post-mortem examination. The main clinical manifestations are endocarditis, aortic graft infection, myocarditis and sternotomy wound infection [3,8,9,10,11,12]. 

Treatment regimens of endovascular MC infection consist of azithromycin, rifampin/(rifabutin), ethambutol, amikacin (first-line therapy) or clarithromycin, rifabutin/(rifampin), ethambutol, amikacin (second-line therapy) [13,14]. Patients’ therapy included several months of antibiotics, and a new surgical intervention appeared to be critical for successful outcomes [8,15,16,17,18,19]. In the cases of disseminated MC infection, several manifestations occurring after initiation of treatment have represented an immune reconstitution inflammatory syndrome (IRIS) including fever and abscess formations in various body sites [9].

Diagnosis of endovascular MC infection can be difficult, as standard diagnostic methods offer varying levels of sensitivity [8,19,20,21,22,23,24]. Moreover, the diagnosis may be elusive due to the long latency between infection and diagnosis, and because of often-intermittent bacteraemia and normal echocardiography in some cases. MC detection in the environment can also be challenging, especially due to the biofilm formation ability of MC that makes disinfection of medical devices and eradication difficult [23,25,26,27,28].

Switzerland was the leading country in recognizing and investigating this global outbreak [29]; at least one hundred eighty cases were reported worldwide, involving over one hundred facilities and five HCU manufacturers [3,4,8,29,30,31,32,33,34,35,36,37,38,39,40,41,42]. The absolute risk of acquiring MC infection has been estimated to be much lower than the risk of other types of infections after open chest surgery (from 0.14/1000 surgical interventions in UK or 0.78/1000 surgical interventions in Switzerland to around 1/10,000 [29,41,42]). However, because of the long latency between infection and the onset of symptoms and microbiological confirmation, many cases remained unrecognized for long periods of time [1,30,43], and it should be presumed that many more cases have gone unnoticed. Several additional risk factors have been reported for MC: cardiopulmonary bypass surgery with/without implantation of foreign material, length of extracorporeal circulation time, type of HCU (Stockert 3T, Maquet, others), HCU position in/outside the operating room and distance from the operating table, HCU disinfection/maintenance status, patients’ degree of immunosuppression [8,31,44,45,46,47]. In some studies, aortic surgery was identified as the procedure carrying the highest risk, while the lowest risk was associated with coronary artery bypass grafting. Infections have also been reported among patients following minimal-access cardiac surgery through small lateral thoracotomies [8,9,44]. The high frequency of surgical reintervention despite antimycobacterial therapy and the occurrence of breakthrough infections were all factors that contributed to the severe outcomes (30 to 50% relapse rate, 20–67% mortality rate) [3,20]. All of the above factors rendered this outbreak, unprecedented because of its global scale, a significant burden for health care systems in terms of extra costs incurred and future medicolegal implications [48].

Following the publication by the European Centre for Disease Prevention and Control (ECDC) of a Rapid Risk Assessment document in 2015, which was updated in 2016 [49], the Italian Ministry of Health issued the guidelines for a surveillance scheme to assess the risk of MC infection. The surveillance aimed to examine retrospectively cases with patients who had undergone chest surgical procedures as far back as 2010, and until December 2022.

In this paper, we report a comprehensive analysis of the HCU-related outbreak occurred in Italy, through a retrospective surveillance covering the years 2010–2022. The data presented include microbiological and molecular analyses of strains isolated from patients as well as from the HCUs used in the hospitals where the surgeries were performed. 

## 2. Materials and Methods

### 2.1. Epidemiological Investigation

Epidemiological surveillance of human infections due to NTMs is mandatory in Italy since 1990 and is regulated by the Italian Ministry of Health. The following case definitions for HCU-related invasive infection by MC were derived from indications received from the ECDC and are based on both clinical and exposure criteria. Of note, respiratory samples were not considered “significant biological samples”.

Suspected case: a patient who met clinical and exposure criteria, for whom no microbiological confirmation was available;Probable case: a patient who met the clinical and exposure criteria and had MC identified by direct PCR and Sanger sequencing in a significant biological sample, or a MAC strain isolated by culture or by direct PCR from a significant biological sample, or histopathological detection of nongaseous granuloma and foamy/swollen macrophages with the presence of alcohol-acid-fast bacilli in cardiac or vascular tissue or in a sternotomy wound specimen;Confirmed case: a patient who met the clinical and exposure criteria and had MC isolated by culture and identified by Sanger sequencing in a significant biological sample [49].

For comparison, 21 strains grown from respiratory samples of patients with no epidemiological link to open chest surgery were included in the analysis.

Nineteen out of twenty Italian regions and both Autonomous Provinces (Trento and Bolzano) were asked by the Italian Ministry of Health to provide all retrospective information, collected from 2010 onwards, for the following sample types: blood, bone marrow, bone biopsy, cerebrospinal fluid, pleural fluid or abdominal effusion, soft tissue, wound or abscess drainage, lymph nodes. Cases were identified on the basis of routine diagnostic results. 

Microbiological investigations on significant biological samples were carried out by the laboratories of the hospitals where the suspected cases were managed, as per the protocol prepared by the Italian Ministry of Health.

The HCU devices were also investigated. The vigilance system regulated by the Legislative Decree 46/97 and European Directive 93/42 requires that the manufacturers and the healthcare professionals notify the relevant Competent Authority of any incidents involving medical devices. Accordingly, all HCU contaminations by MC had to be reported to the Italian Ministry of Health. When identified, cases infected by *M. chimaera*, including those without a history of HCU-assisted surgery, were mandatorily reported to the Italian Ministry of Health within 7 days of microbiological confirmation, and subsequently updated with information on evolution and outcome, with a 12-month follow-up period [50]. Data were analysed by the statistical software R, version 4.1.1, stratified by sex and geographical area according to the Italian National Institute of Statistics (ISTAT, https://www.istat.it/it/archivio/240401, accessed on 16 January 2023; demographical data).

### 2.2. Molecular Analysis of Strains

The National Institute for Infectious Diseases “L. Spallanzani”-IRCCS (INMI) in Rome was appointed by the Italian Ministry of Health as the National Reference Laboratory for the genomic sequencing and molecular/epidemiological analysis of the outbreak-related strains. *M. chimaera* strains sent to INMI were subcultured in MGIT medium (MGIT, Becton Dickinson, MD, USA), and genomic DNA was extracted according to the manufacturer’s protocol using QIAamp DNA minikit (Qiagen, Hilden, Germany). Before sequencing, DNA was quantified by Qubit 4.0 using the Qubit dsDNA HS assay kit (Thermo Fisher Scientific, Waltham, MA, USA) and subsequently processed, as described in the manufacturer’s protocol for whole genome sequencing (WGS), using the Ion Xpress Plus Fragment Library Kit (Thermo Fisher Scientific, USA) for sample library preparation. The Chef and S5 platforms were used for automated chip preparation and sequencing, respectively; the procedure resulted in the production of 250 bp reads. The quality of readings was evaluated using FastQC software v0.11.9 and only the fastq files with high-quality standard were analysed [51]. The reference genome DSM-44623 sequence (NZ CP015278.1) was used for mapping process by BWA v0.7.17-r1188 and samtools v1.9 [4,52]. In fact, bioinformatic analysis was carried out using a pipeline that was specifically developed for reconstruction of MC genome [52], where bwa mem made it possible to perform an alignment processing and samtools functions (fixmate, sort and index) that provided bam files for the variant calling step. Variant calling was performed with freebayes v1.3.2 [53]. Finally, genomic variability variants in *M. chimaera* genomes were obtained, after imposing haploid genome mode, only SNPs, 5-fold minimum coverage, 90% allele frequency, 50 mapping quality and 30 base quality (-F 0.9 -p 1 -i -X -u -m 50 -q 30 min coverage 5). The group and subgroup classification of strains was carried out according to the method described by van Ingen and colleagues [4].

## 3. Results

### 3.1. Demographic and Statistical Analysis

Through classical epidemiology, 40 possible cases of HCU-related MC infection were reported to the Italian Ministry of Health for the study period (January 2010–December 2022) (Figure 1). 

The first case was identified in 2017, while retrospective investigations allowed us to identify cases dating back to 2015 (Table 1). The cases were reported from 6/20 regions (Veneto 23, Emilia-Romagna 7, Piedmont 5, Calabria 3, Friuli Venezia-Giulia 1 and Lombardy 1). Twenty-two deaths were recorded (mortality rate 55.0%). Most (87.5%) of the cases were male, with a mean age of 64.9 years (median age 66, range 35–82 years), and were reported from Northern Italy (92.5%). Mostly, patients showed fever (10.8%), leukopenia/thrombocytopenia (9.5%), endocarditis (8.1%), hepatitis (8.1%), asthenia (6.8%), weight loss (6.8%) and night sweats (6.8%). The mean time between surgery and diagnosis was 4.6 years, with a latency of missed diagnosis ranging between 2 and 8 years. Death occurred between 0.5 and 9.5 years from the onset of symptoms.

### 3.2. Laboratory Investigations

The identification of *M. chimaera* isolates from clinical/environmental samples followed the recommendations issued by the Italian Ministry of Health and according to the International Society of Cardiovascular Infectious Diseases Guidelines [54]; investigations were performed according to the flow diagram presented in Figure 2. 

The molecular epidemiological analysis was performed by WGS on 57 strains isolated from biological samples, collected for this study (Table 2). Such genomic analysis confirmed their belonging to MC species. Among the analysed isolates, 36 (Table 2) had an epidemiological link to the use of HCUs, and were all classified into subgroups 1.1 or 1.8, which are the two subgroups associated with the HCU-related outbreak [4]. Twenty-one strains obtained from respiratory samples of patients with no history of cardiac surgery were also analysed (Table 2): none belonged to subgroups 1.1 or 1.8.

### 3.3. Contaminated HCU Analysis

During the study period, we also identified 251 *M. chimaera*-contaminated HCUs in Italy. In fact, most of the culture tests performed on the HCUs produced by LivaNova (formerly Sorin, London, UK) before September 2014 turned out positive for MC. Moreover, in some cases the *Mycobacterium* was isolated after deep decontamination of the devices at the manufacturer site. In 2017, the Italian Society of Cardiac Surgery launched a national survey among the Italian ACSU to shed some light on this issue [16]. 

These contaminations were found in Northern (87.1%) and Central (12%) Italy, and Islands (0.9%). No notifications were registered from Southern Italy. In particular, these 251 contaminated HCUs were reported from 10 regions (Emilia-Romagna, Friuli Venezia-Giulia, Lombardy, Piedmont, Tuscany, Umbria, Veneto, Lazio, Sardinia and Sicily). A total of 87 MC isolates from contaminated HCUs were also sequenced by WGS, resulting in 28 isolates found to belong to 1.1 or 1.8 subgroups (Table 3).

## 4. Discussion

### 4.1. Public Health Response to Outbreak of MC Invasive Infection during Open Chest Surgery Occurred in Italy

The first Italian case of MC disseminated infection was identified in 2017, when Chiesi and colleagues (2017) reported a case of a 70-year-old woman who had undergone bio prosthetic mitral valve replacement in 2014; the patient had initially been diagnosed with fever (37.5 °C), cough, fatigue and disseminated sarcoidosis [55].

Forty cases were identified in total (Table 1); the patients were mainly males (87.5%) and predominantly from the northern part of the country. The higher percentage of male patients is likely due to the higher prevalence in males of the underlying conditions that led these patients to cardiac surgery [56], while the higher concentration of cases in northern Italy likely reflects the geographical diversity of a highly decentralized health system [57]. As occurred in other countries, also in Italy, the detection of cases, as well as their contamination source, has been challenging, because of the already mentioned long latency of symptoms onset and because the symptoms can be nonspecific and an epidemiological link may not be immediately clear. As an aid to clinicians, the International Society for Cardiovascular Infectious Diseases (ISCVID) published guidelines covering aspects of prevention, clinical management, laboratory diagnostics and public health aspects [54].

In June 2019, the Italian Ministry of Health established a national expert working group comprising infectious disease specialists, cardiac surgeons, microbiologists, perfusion technicians, public health experts, pharmacists, in order to investigate the outbreak (June 2019, Ministerial Decree) [50]. 

Overall, three circular letters were sent out in January, April and November 2019 and addressed to all twenty regions (R) and two autonomous provinces (AP). The letters contained operational guidelines for case management, laboratory diagnosis, recommendations for preventing further HCU contamination and human infections, as well as updates on the number of cases and contaminated HCUs [58]. In April 2019, the Italian Ministry of Health identified the L. Spallanzani institute as the National Reference Laboratory for molecular epidemiology investigations. Local laboratories were responsible for microbiological identification, while the national reference centre would carry out WGS analysis to confirm the association with the outbreak. The Italian Ministry of Health surveyed every HCU used in the country and urged the R/AP to implement appropriate measures to ensure the following:Identification of the subjects exposed by setting up a register of patients who had undergone HCU-assisted open chest surgery (including cardiac and/or pulmonary transplantation and aortic vascular transplantation) at the hospital level;Implementation of a local register of the HCUs used (where lacking), at the hospital level, in order to allow fast retrospective identification of patients at risk;Setting up regional/interregional reference centres for advice to medical staff and exposed subjects and follow-up of cases;Identification, where possible, of the general practitioners who cared for subjects potentially at risk to send them information for patients’ follow up (in addition to ministerial circular letters and any regional documents);Withdrawal of devices associated with one or more cases until microbiological proof of noncontamination.

Overall, the manufacturer LivaNova implemented seven field safety corrective actions as required by the current EU Legislation on medical devices, aimed at reducing the risk of infection. In May 2017, a safety notice recommended a retrofit for type 3 HCU machines and decommissioning of type 1 HCU machines. In 2021, Quintas Viqueira and colleagues showed that the 3T heater–cooler devices placed outside of the operating room (as per manufacturer’s instructions) resulted the best measure to prevent MC infection during cardiac surgery [34].

The hospitals needed to comply with the identified corrective actions (decontamination of their HCUs or replacement of the HCU 1T with new devices, or use of the “retrofit” tray provided by LivaNova for the HCU 3T to collect the aerosol generated from the devices). Moreover, use of sterile water was recommended, besides positioning the HCU outside of the operating room [59]. Training of health care professionals was advised, and patients had to be given information regarding MC and its limited ability of causing clinically relevant infections under normal health conditions. Instead, patients who had undergone open-heart surgery requiring extracorporeal blood circulation were advised to inform their doctor about the surgery performed and, if applicable, any symptoms. It was the doctor’s responsibility to enquire with the hospital where the patient underwent the surgery and to assess the risk of a possible infection with MC.

It was also recommended that, if possible, molecular epidemiological analysis should be performed on strains isolated from both cases and HCUs. 

### 4.2. Operating Procedures to Contain the Spread of Infections

According to the results from the National Survey endorsed by the Italian Society of Cardiac Surgery, the risk for MC infections was 0.4–1.0 patient every 1000 cardiac procedures [16]. It was recognized that systematic screening of patients rather than individually based screening of persons considered at risk might provide a higher number of early diagnoses. However, given the long latency (generally between 2 and 8 years) and the relatively low negative predictive value of diagnostic methods for MC, patients should be followed up and reassessed periodically for years. 

Given that the use of HCUs has been identified as the source of contaminated aerosol during open chest surgical procedures, the correct cleaning, disinfection and maintenance of such devices are key routine actions to guarantee patient safety. Adhering to a strict maintenance schedule according to the manufacturer’s instructions for use (IFU) is essential to reduce the risk of infection. The effectiveness of these interventions should be monitored periodically. 

Based on information provided by the manufacturer and scientific evidence, the Italian Ministry of Health took steps to ensure that all information was adequately disclosed and carried out a detailed reconnaissance to locate the 1T machines, ascertaining that every device had been decommissioned. The Italian Ministry of Health continued to monitor the implementation of the corrective actions recommended by LivaNova; by March 2019, all 156 HCU 3T (number provided by the manufacturer) were retrofitted as instructed by the manufacturer. 

Any new information on HCU contamination must be reported to the Italian Ministry of Health as per the current legislation concerning vigilance on medical devices. 

## 5. Limitations of this Study

One important limitation of our study is that it was based on cases reported to the Italian Ministry of Health and/or whose samples were sent to the National Reference Laboratory for molecular testing. Italy has a “federal” national health system in which every region (20 in total) has a high degree of autonomy and the powers and responsibilities are highly decentralized [57]. Although surveillance and notification of cases was mandatory, this peculiarity of the Italian health system has almost certainly produced a geographical bias, whereby some regions may have been more effective than others in screening for cases and may explain why cases were reported from Northern (92.5%) and Southern (7.5%) Italy only. 

The risk of underestimating MC infection is also discussed in the literature [29,36,60] so it can be presumed that many cases might have been missed. The Italian Ministry of Health has stressed the need for accurate case notification as well as for regular monitoring of all patients who underwent open chest surgery before the use of the Retrofit system in 3T machines and all those operated on 1T machines.

## 6. Conclusions

There are lessons to be learned from this global outbreak of invasive MC infections correlated with the use of HCU medical devices. Further research on many aspects of diagnosis, management and prevention is certainly needed, such as assessing the risk for the paediatric population, developing effective decontamination procedures to disrupt biofilms and improving the design of HCUs in order to facilitate decontamination. Although the role of other mycobacteria, fungi, *Legionella* spp., and other pathogens in HCU-associated infections should be studied, but certainly a regular and effective surveillance of HCUs is the main preventive measure to adopt [61].

A better integration of molecular investigations with epidemiological and clinical data would also be extremely beneficial for future outbreak investigations. Due to the rarity of the disease, multicentre data collections would allow for consistency of evidence-based decisions regarding epidemiology, clinical manifestations, treatment and outcomes. In the Italian context, this experience should teach us to increase communication between all actors at local, regional and central levels to improve case detection and notification (regarding cases and HCUs) in order to put in place effective health policies in the future and overcome the potential geographical weaknesses of a decentralized health system.

Another important aspect to consider is public awareness and education, as well as education of health care providers to improve their communication skills in dealing with patients who are likely to develop psychological issues caused by the long follow-up. 

Ten years after the first cases were reported worldwide, and although all the above-described corrective actions have been implemented, we cannot say that the emergency is over, because due to the long latency between exposure and disease, we can expect that more cases may be uncovered in the near future. 

This global outbreak represented an opportunity to reinforce awareness on the correct use of HCUs, the importance of cutting-edge genomic methods applied to molecular and timely epidemiological surveillance, as well as prompt communication of cross-border infectious diseases.

## Figures and Tables

**Figure 1 microorganisms-11-02251-f001:**
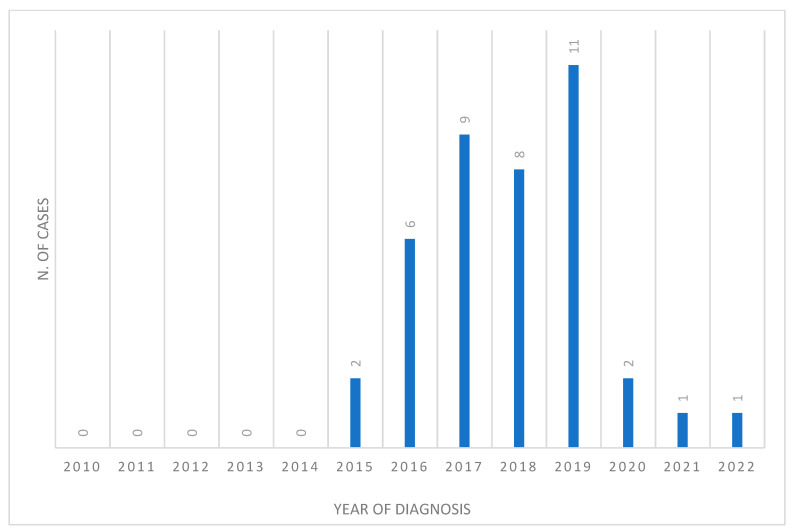
Epidemic curve showing the number of MC cases retrospectively reported to the Italian Ministry of Health, by year of diagnosis, Italy, January 2010–December 2022 (n = 40).

**Figure 2 microorganisms-11-02251-f002:**
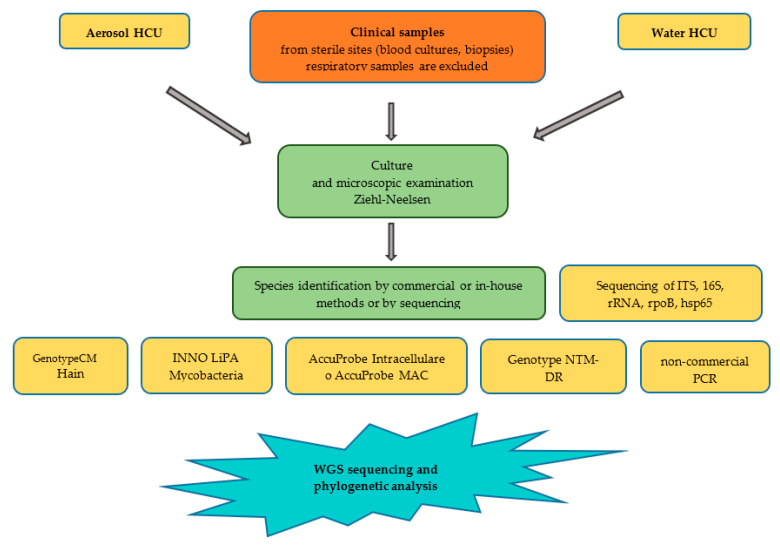
Flow diagram used to identify and characterize MC isolates from clinical/environmental samples.

**Table 1 microorganisms-11-02251-t001:** Demographic and clinical characteristics of cases infected by *M. chimaera*, Italy, January 2010–December 2022 (n = 40).

Total Cases (n = 40)	Absolute Number (%)
Demographics and risk factors	
Male	35 (87.5)
Geographical area: Northern Italy	37 (92.5)
Geographical area: Central Italy	0
Geographical area: Southern Italy	3 (7.5)
Geographical area: Islands	0
Main symptoms (n = 14)	
Fever	8 (10.8)
Leukopenia/thrombocytopenia	7 (9.5)
Endocarditis	6 (8.1)
Hepatitis	6 (8.1)
Asthenia	5 (6.8)
Weight loss	5 (6.8)
Night sweats	5 (6.8)
Multiple granulomas	4 (5.4)
Nephritis	4 (5.4)
Lung involvement/pneumonia	4 (5.4)
Splenomegaly	4 (5.4)
Nausea	3 (4.1)
Bacteraemia	2 (2.7)
Death	21 (52.5)
Mean; median age (range)	64.9; 66 (35–82 years)
Mean; median time between surgery and diagnosis (range, n = 37)	4.6; 4 (2–8 years)
Mean; median time between diagnosis and death (range, n = 16)	2.1; 1.5 (0.5–9.5 years)

**Table 2 microorganisms-11-02251-t002:** Informative SNPs and classification of M. chimaera strains isolated from patients undergoing cardiac surgery with use of heater–cooler units (HCUs), Italy, January 2010 to December 2022. These data were obtained by high-throughput sequencing (NGS) of whole genome. (**A**) Patients with an epidemiological link, defined as having had cardiac surgery with use of heater–cooler units (HCUs), Italy, January 2010 to December 2022. (**B**) Patients without an epidemiological link, i.e., who were not subjected to cardiac surgery with use of heater–cooler units (HCUs).

(A)		
ID	SNPs [4]	Classification
1	113518G>A; 209278G>A	1.1
2	113518G>A; 209278G>A	1.1
3	113518G>A; 209278G>A	1.1
4	1611282G>C; 2366314G>A	1.8
5	113518G>A; 209278G>A	1.1
6	113518G>A; 209278G>A	1.1
7	113518G>A; 209278G>A	1.1
8	113518G>A; 209278G>A	1.1
9	113518G>A; 209278G>A	1.1
10	113518G>A; 209278G>A	1.1
11	113518G>A; 209278G>A	1.1
12	113518G>A; 209278G>A	1.1
13	113518G>A; 209278G>A	1.1
14	113518G>A; 209278G>A	1.1
15	1611282G>C; 2366314G>A	1.8
16	1611282G>C; 2366314G>A	1.8
17	113518G>A; 209278G>A	1.1
18	113518G>A; 209278G>A	1.1
19	1611282G>C; 2366314G>A	1.8
20	113518G>A; 209278G>A	1.1
21	113518G>A; 209278G>A	1.1
22	113518G>A; 209278G>A	1.1
23	1611282G>C; 2366314G>A	1.8
24	113518G>A; 209278G>A	1.1
25	113518G>A; 209278G>A	1.1
26	113518G>A; 209278G>A	1.1
27	113518G>A; 209278G>A	1.1
28	113518G>A; 209278G>A	1.1
29	113518G>A; 209278G>A	1.1
30	113518G>A; 209278G>A	1.1
31	113518G>A; 209278G>A	1.1
32	113518G>A; 209278G>A	1.1
33	113518G>A; 209278G>A	1.1
34	113518G>A; 209278G>A	1.1
35	113518G>A; 209278G>A	1.1
36	1611282G>C; 2366314G>A	1.8
**(B)**		
**ID**	**SNPs [4]**	**Classification**
1	4977262T>C	ungrouped
2	4977262T>C	ungrouped
3		ungrouped
4		ungrouped
5	5003561A>G	Branch 2
6	4977262T>C	Branch 1
7	4977262T>C	Branch 1
8		ungrouped
9	5003561A>G	Branch 2
10		ungrouped
11		ungrouped
12	4977262T>C	Branch 1
13		ungrouped
14	5003561A>G	Branch 2
15		*Roseomonas mucosa*
16		ungrouped
17	4977262T>C	Branch 1
18	4977262T>C	Branch 1
19	4977262T>C	Branch 1
20	4977262T>C	Branch 1
21		ungrouped

**Table 3 microorganisms-11-02251-t003:** Informative SNPs and classification of MC strains isolated from heater–cooler units (HCUs) used for patients who had undergone cardiac surgery, Italy, January 2010 to December 2022. These data were obtained by high-throughput sequencing (NGS) of whole genome.

ID	SNPs [4]	Classification
1	3022332T>C; 5709901T>C; 3406341C>T; 1828053C>T	2
2	113518G>A; 209278G>A	1.1
3	113518G>A; 209278G>A	1.1
4		ungrouped
5		ungrouped
6		ungrouped
7	113518G>A; 209278G>A	1.1
8	113518G>A; 209278G>A	1.1
9	5003561A>G; 2339764C>T	Branch 2
10	5003561A>G	Branch 2
11	5003561A>G; 2339764C>T	Branch 2
12	5003561A>G; 2339764C>T	Branch 2
13	113518G>A; 209278G>A	1.1
14	113518G>A; 209278G>A	1.1
15	113518G>A; 209278G>A	1.1
16	113518G>A; 209278G>A	1.1
17	113518G>A; 209278G>A	1.1
18		ungrouped
19	113518G>A; 209278G>A	1.1
20	4977262T>C	Branch 2
21		ungrouped
22		ungrouped
23	113518G>A; 209278G>A	1.1
24		ungrouped
25	4977262T>C	Branch 2
26		ungrouped
27	4977262T>C	Branch 1
28		ungrouped
29	113518G>A; 209278G>A	1.1
30		ungrouped
31	4977262T>C	Branch 1
32	113518G>A; 209278G>A	1.1
33		*Sphingomonas paucimobilis*
34	113518G>A; 209278G>A	1.1
35	4977262T>C	Branch 1
36	4977262T>C	Branch 1
37		*Oligotropha carboxidovorans*
38	4977262T>C	Branch 1
39		*M. paragordonae*
40	113518G>A; 209278G>A	1.1
41		Branch 2
42		ungrouped
43		ungrouped
44	113518G>A; 209278G>A	1.1
45		ungrouped
46		ungrouped
47		ungrouped
48	113518G>A; 209278G>A	1.1
49	3022332T>C; 3406341C>T; 1828053C>T	2.1
50	5003561A>G	Branch 2
51	5003561A>G	Branch 2
52	5003561A>G	Branch 2
53	5003561A>G	Branch 2
54	5003561A>G	Branch 2
55	5003561A>G	Branch 2
56	5003561A>G	Branch 2
57	5003561A>G	Branch 2
58	113518G>A; 209278G>A	1.1
59	113518G>A; 209278G>A	1.1
60		ungrouped
61		ungrouped
62		ungrouped
63	2339764C>T	Branch 2
64		ungrouped
65		ungrouped
66		ungrouped
67		ungrouped
68		ungrouped
69		ungrouped
70		ungrouped
71	3406341C>T; 1828053C>T; 2329494C>T	2.1
72	1611282G>C; 2366314G>A	1.8
73	3022332T>C; 3406341C>T; 1828053C>T	2.1
74	113518G>A; 209278G>A	1.1
75	113518G>A; 209278G>A	1.1
76	5003561A>G	Branch 2
77	5003561A>G	Branch 2
78	5003561A>G	Branch 2
79	113518G>A; 209278G>A	1.1
80	2329494C>T; 5709901T>C	2
81	113518G>A; 209278G>A	1.1
82	113518G>A; 209278G>A	1.1
83	113518G>A; 209278G>A	1.1
84	113518G>A; 209278G>A	1.1
85	2329494T>C; 3022332T>C; 3949608A>G; 5709901T>C	2
86	4977262T>C	Branch 1
87	113518G>A; 209278G>A	1.1

## Data Availability

All of the data generated or analysed during this study are included in this published article and are available upon reasonable request to the corresponding author.

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
