# Peer review of "Epidemiological and Molecular Investigation of the Heater–Cooler Unit (HCU)-Related Outbreak of Invasive Mycobacterium chimaera Infection Occurred in Italy"

_microorganisms, 2023, doi:10.3390/microorganisms11092251_

Round 1

Reviewer 1 Report

From the point of view of practical and clinical microbiology, the manuscript is a valuable contribution to the assessment of M. chimaera as a clinically important nontuberculous species of Mycobacterium. Its value also lies in the collection of clinical and microbiological data from the territory of Italy. It is advisable to add an epidemiological map of Italy to the text, indicating the monitored areas and the number of cases in individual areas.

However, the manuscript primarily needs formal editing. Many sentences are too long. The text must be divided into several paragraphs to increase its clarity and readability. The text is of professional and scientific quality. However, large paragraphs should be divided into smaller paragraphs with a maximum of 10-15 lines. This will make the text more readable.

Lines 67-72, Lines 126-134: Sentences are not clear, please, re-write this parts of the text.

New paragraphs must be created (Lines are in blue in original manuscript): Lines 64, 72, 82, 108, 116, 134, 138, 153, 167.

English needs checking by native speakers.

Specific comments

Line 29: add „respectively“.

Line 37: from keywords remove „Mycobacterium chimaera“. This name of the pathogen is already in the title.

Line 42 and Tab. 3 ID 39: use abbreviated name of the Genus Mycobacterium as „M.

Line 109: The generic name must be given in the form (italic): Mycobacterium

Lines 363, 364, 377, 404, 407, 419, 427,437, 449, 572Latin names must be in italic

Line 134: Please, add map of Italy with regions and Autonomous Provinces with the number of reported cases

Line 217: Table 3: Remove column “Sample” with repeated HCU which is in the title of the table.

Line 248: Which ministry? Please, specify.

Line 105: Abbreviation IT-MOH use thorough the whole text or remove it on this line and line 347.

Line 212: Table 2. Strains sort, please, in two categories: with and without epidemiological link.

Line 233: Discussion. Try to explain, why females were not so often diagnosed as infected patients.

Line 288: numbers of patients write in this form: 0.4–1.0

English needs checking by native speakers.

Author Response

From the point of view of practical and clinical microbiology, the manuscript is a valuable contribution to the assessment of M. chimaera as a clinically important nontuberculous species of Mycobacterium. Its value also lies in the collection of clinical and microbiological data from the territory of Italy. It is advisable to add an epidemiological map of Italy to the text, indicating the monitored areas and the number of cases in individual areas.

However, the manuscript primarily needs formal editing. Many sentences are too long. The text must be divided into several paragraphs to increase its clarity and readability. The text is of professional and scientific quality. However, large paragraphs should be divided into smaller paragraphs with a maximum of 10-15 lines. This will make the text more readable.

Response: we have re-written large parts of the article, breaking down sentences that were too long.

We gave careful consideration to the suggestion of adding an epidemiological map of Italy, with monitored areas and number of cases. It is a good suggestion, however we believe that, given the peculiarity of the Italian situation, it would not reflect reality. The Italian health system is highly decentralized, to the point that it can be considered a ‘federal’ system, where each Region is autonomous in many respects. A consequence of this is that it may well be that some Regions were more efficient in case finding and follow up. Moreover, the hospitals in the northern regions of the country are in general better organized and more efficient, and many patients from the centre or the south travel north to be treated especially for serious diseases and surgery, so the number of cases would not reflect the local epidemiology. We have included this information in the text and discussed this geographical bias as a limitation of the study in a new paragraph (Page 12, lines 339-344) We also added reference 57 that describes the organization of the health service in Italy.

Lines 67-72, Lines 126-134: Sentences are not clear, please, re-write this parts of the text.

Response: we have re-written the sentences that were unclear, and re-phrased as suggested also by Reviewer 2 (page 3, lines 98-102, 105-107, and page 2, lines 82-85).

New paragraphs must be created (Lines are in blue in original manuscript): Lines 64, 72, 82, 108, 116, 134, 138, 153, 167.

Response: we have created new paragraphs throughout the paper

English needs checking by native speakers.

Response: the paper was checked by a native speaker

Specific comments

Line 29: add „respectively“.

Response: the abstract has been completely re-written, separating data obtained from patients and HCU

Line 37: from keywords remove „Mycobacterium chimaera“. This name of the pathogen is already in the title.

Response: we removed M. chimaera from keywords

Line 42 and Tab. 3 ID 39: use abbreviated name of the Genus Mycobacterium as „M.“

Response: Done

Line 109: The generic name must be given in the form (italic): Mycobacterium

Response: Done (now line 226)

Lines 363, 364, 377, 404, 407, 419, 427,437, 449, 572: Latin names must be in italic

Response: Done

Line 134: Please, add map of Italy with regions and Autonomous Provinces with the number of reported cases

Response: please see  our response above, where we explain the geographical bias of the Italian health system. We hope the reviewer will find this explanation acceptable.  We have mentioned  this “geographical bias” as a limitation of the study, and also a lesson learned for the future (Page 12, lines 339-344 and page 13, lines 365-369).

Line 217: Table 3: Remove column “Sample” with repeated HCU which is in the title of the table.

Response: Done

Line 248: Which ministry? Please, specify.

Response: Done, we specified it was the Ministry of Health, throughout the paper

Line 105: Abbreviation IT-MOH use thorough the whole text or remove it on this line and line 347.

Response: Done, we removed the abbreviation

Line 212: Table 2. Strains sort, please, in two categories: with and without epidemiological link.

Response: Done, now Table 2 is composed of Table 2A and Table 2B

Line 233: Discussion. Try to explain, why females were not so often diagnosed as infected patients.

Response: we believe that this is a reflection of the underlying condition that led the patients to cardiac surgey that is responsible for this discrepancy (Page 11, lines 252-254, and reference 56)

Line 288: numbers of patients write in this form: 0.4–1.0

Response: Done (page 12, line 313).

Comments on the Quality of English Language

English needs checking by native speakers.

Response: the English was checked by a native speaker.

Reviewer 2 Report

The manuscript provides a comprehensive look at the Mycobacterium chimaera outbreak in Italy associated with HCU devices. However, it requires revision for clarity, consistency, and format. Consider obtaining and incorporating feedback from clinicians or experts in epidemiology and molecular biology for additional insights specific to the field.

General Comments:

The title suggests that the study focuses on the outbreak between 2020 and 2022, but the manuscript mentions data collection and cases from as early as 2010. This discrepancy should be clarified.

Abstract:

1.      The sentence structure of "As occurred in other countries, Italy was involved too in outbreak..." could be improved for clarity.

2.      The sentence "Most of the cases presented with prosthetic valve endocarditis..." contains results. If you want to maintain a traditional IMRaD (Introduction, Methods, Results, and Discussion) format, consider moving such details to a results section.

Introduction:

1.      The introduction provides extensive details, some of which may be better suited for a background or literature review section. Consider providing a clear objective or research question at the end of the introduction to guide the reader on the purpose of the study.

2.      It's mentioned that Mycobacterium chimaera was identified in 2004, but a reference for this claim is missing.

3.      Ref [8] seems to be referenced multiple times. Please ensure that this is a comprehensive source that covers all the cited details.

Materials and Methods:

1.      The epidemiological case definition is comprehensive but could benefit from bullet points or another clear formatting choice to differentiate between confirmed and probable cases.

2.      This section could also provide more details on the selection criteria for the cases and controls if any were included.

3.      In molecular analysis, the methods used for genomic sequencing, data processing, and analysis need to be expounded upon for clarity and reproducibility.

4.      Please consider adding details on the ethical considerations and approvals for the study.

Results:

It would be helpful to have a brief explanation for readers unfamiliar with HCU and its relation to M. chimaera infections. The importance of this study lies in understanding this context.

Discussion:

The Operating Procedures section does a good job of emphasizing the importance of systematic screening and the challenges that come with it. However, discussing the implications of these findings on future health policies in Italy would further strengthen the discussion.

Conclusion:

The emphasis on the need for collaboration at various levels is commendable. It might also be beneficial to stress the importance of public awareness and education, given the challenges in early diagnosis and the long latency of symptoms.

The English in the manuscript is mostly clear and comprehensible. However, there are a few areas that might benefit from some minor revisions for clarity or improved grammar. I recommend a moderate level of editing to refine the English language and ensure that the research is presented as clearly as possible.

Author Response

Reviewer 2

The manuscript provides a comprehensive look at the Mycobacterium chimaera outbreak in Italy associated with HCU devices. However, it requires revision for clarity, consistency, and format. Consider obtaining and incorporating feedback from clinicians or experts in epidemiology and molecular biology for additional insights specific to the field.

General Comments:

The title suggests that the study focuses on the outbreak between 2020 and 2022, but the manuscript mentions data collection and cases from as early as 2010. This discrepancy should be clarified.

Response: we have clarified that the surveillance was retrospective, so although cases were internationally reported in later years, the Italian Ministry of Health decided to analyse records of cases dating as far back as 2010, until December 2021. We clarified this throughout the paper, for example Page 3, lines 108-109. To avoid any confusion, we modified the title, removing the phrase ‘between 2020 and 2022’

Abstract:

The sentence structure of "As occurred in other countries, Italy was involved too in outbreak..." could be improved for clarity.

Response: the whole abstract has been re-written to improve clarity

The sentence "Most of the cases presented with prosthetic valve endocarditis..." contains results. If you want to maintain a traditional IMRaD (Introduction, Methods, Results, and Discussion) format, consider moving such details to a results section.

Response: we have extensively re-written the article to maintain a traditional IMRaD structure

Introduction:

The introduction provides extensive details, some of which may be better suited for a background or literature review section. Consider providing a clear objective or research question at the end of the introduction to guide the reader on the purpose of the study.

Response: we have re-written the introduction to improve readability, and we added a clear aim of the study (page 3, lines 110-113).

It's mentioned that Mycobacterium chimaera was identified in 2004, but a reference for this claim is missing.

Response: we linked with reference n. 1

Ref [8] seems to be referenced multiple times. Please ensure that this is a comprehensive source that covers all the cited details.

Response: we checked all the references, including ref. 8. In the revised paper, as a result of the re-organization of paragraphs, also the reference list has been checked and updated.

Materials and Methods:

The epidemiological case definition is comprehensive but could benefit from bullet points or another clear formatting choice to differentiate between confirmed and probable cases.

Response: we have included bullet points, and re-written the sentences that were unclear, as suggested also by Reviewer 1 (page 3, lines 121-136).

This section could also provide more details on the selection criteria for the cases and controls if any were included.

Response: in addition to the case definition, we added a sentence specifying how the ‘control’ strains (i.e. those from patients with no epidemiological link to cardiac surgery) were included in the study (Page 3, lines 133-134).

In molecular analysis, the methods used for genomic sequencing, data processing, and analysis need to be expounded upon for clarity and reproducibility.

Response: We added a sentence to provide additional information, plus two new references (Page 4, lines 170-177, references 52 and 53).

Please consider adding details on the ethical considerations and approvals for the study.

Response: we added a paragraph entitled ‘ethical considerations’: Page 13, lines 382-388).

Results:

It would be helpful to have a brief explanation for readers unfamiliar with HCU and its relation to M. chimaera infections. The importance of this study lies in understanding this context.

Response: we added a paragraph that gives a background to understand the problem with HCU (Page 2, lines 50-55). References 4 and 5 are also very comprehensive in the description of the problem.

Discussion:

The Operating Procedures section does a good job of emphasizing the importance of systematic screening and the challenges that come with it. However, discussing the implications of these findings on future health policies in Italy would further strengthen the discussion.

Response: we added some ‘lessons learned’ and discussed the weaknesses identified (Page 13, lines 353-376).

Conclusion:

The emphasis on the need for collaboration at various levels is commendable. It might also be beneficial to stress the importance of public awareness and education, given the challenges in early diagnosis and the long latency of symptoms.

Response: we added a paragraph on the importance of public awareness and education, not only of the public, but also of the health care professionals dealing with the public (Page 13, lines 370-372).

Comments on the Quality of English Language

The English in the manuscript is mostly clear and comprehensible. However, there are a few areas that might benefit from some minor revisions for clarity or improved grammar. I recommend a moderate level of editing to refine the English language and ensure that the research is presented as clearly as possible.

Response: we have checked the manuscript for clarity and grammar

Round 2

Reviewer 1 Report

General comments

The quality of the manuscript improved greatly after the edits that respected the opponent's suggestions. The non-delivery of the epidemiological map of Italy and the occurrence of M. chimaera was surprisingly but very well described, explained and discussed. The quality of English has been significantly improved.

Minor revisions

Lin 42: rewrite M. Avium Complex to M. avium complex

Line 177: M. chimaera in italic, please

Line 328: „Ministry“ re-write: Italian Ministry of Health

Line 332: „ Ministry of Health“ re-write: Italian Ministry of Health

Author Response

General comments

The quality of the manuscript improved greatly after the edits that respected the opponent's suggestions. The non-delivery of the epidemiological map of Italy and the occurrence of M. chimaera was surprisingly but very well described, explained and discussed. The quality of English has been significantly improved.

Minor revisions

Lin 42: rewrite M. Avium Complex to M. avium complex

Response: Done

Line 177: M. chimaera in italic, please

Response: Done

Line 328: „Ministry“ re-write: Italian Ministry of Health

Response: Done

Line 332: „ Ministry of Health“ re-write: Italian Ministry of Health

Response: Done

All new changes are highlighted in green

Reviewer 2 Report

The revised manuscript details the spread, epidemiology, and control measures of M. chimaera infections related to HCUs in Italy. The overall flow of the article is commendable. The results section is well-detailed and provides a comprehensive insight into the topic.

Please ensure that the manuscript undergoes thorough proofreading to eliminate any minor typographical or grammatical errors that might have been overlooked.

The study sheds light on a critical issue and offers valuable insights into the epidemiology and control measures of M. chimaera infections in Italy. Given its overall merit, I look forward to seeing the work published and making a valuable contribution to the field.

Author Response

The revised manuscript details the spread, epidemiology, and control measures of M. chimaera infections related to HCUs in Italy. The overall flow of the article is commendable. The results section is well-detailed and provides a comprehensive insight into the topic.

Please ensure that the manuscript undergoes thorough proofreading to eliminate any minor typographical or grammatical errors that might have been overlooked.

Response: we checked the manuscript and corrected some typographical errors and inconsistencies. The new changes are highlighted in green

The study sheds light on a critical issue and offers valuable insights into the epidemiology and control measures of M. chimaera infections in Italy. Given its overall merit, I look forward to seeing the work published and making a valuable contribution to the field.